# Work time allocation at primary health care level in two regions of Albania

**Altiona Muho**[1,2]*, **Altina Peshkatari**[3], **Kaspar Wyss**[1,2]

**1** Swiss Centre for International Health, Swiss Tropical and Public Health Institute, Allschwil, Switzerland,
**2** University of Basel, Basel, Switzerland, **3** Health for All Project, Tirana, Albania

* altiona.muho@swisstph.ch

## Abstract

### Introduction

Although well-performing workforce is essential to equitable and efficient health service delivery, few countries have systematically addressed performance improvements. How health workers use their work time and what tasks they accomplish is here an important starting point. Therefore, a time motion study was conducted to assess the work time allocation patterns of primary health care doctors and nurses in two regions of Albania.

### Methods

We used observation tool to record the time allocation along eight predefined main categories of activities. Conditional to presence at work, 48 health workers were continuously observed in early 2020 before start of the Covid-19 pandemic over five consecutive working days.

### Results

The observed health workers spent 40.7% of their overall working time unproductively (36.8% on waiting for patients and 3.9% on breaks), 25.3% on service provision to users, 18.7% on administrative activities, 12.7% on outreach activities, 1.6% on continuous medical education and 1% on meetings. The study found variations in work time allocation patterns across cadres, with nurses spending more time unproductively, on administrative activities and on outreach and less on all other activities than doctors. Further, the work time allocation patterns were similar between urban and rural settings, except for nurses in rural settings spending less time than those in urban settings on administrative work.

### Conclusion

This study found that primary health care workers in Albania devote a substantial amount of work time to unproductive, service provision to users and administrative activities. Consequently, there is possibility for productivity, respectively efficiency gains in how health workers use their time.

**Data Availability Statement:** Data cannot be shared publicly as it contains information that could compromise the privacy of research participants. Data are available upon request from

the authors (contact via the corresponding author and Dr. Kaspar Wyss, Deputy Head of Swiss Tropical and Public Health Institute, kaspar.wyss@swisstph.ch) or from non-author point of contact (via Malin Ziehmer-Wenz, malin.ziehmer-wenz@swisstph.ch).

**Funding:** The study has been conducted in the frame of the Health for all Projekt (Projekti HAP http://www.hap.org.al/) funded by the Swiss Agency for Development and Cooperation (SDC) and implemented by the Swiss Tropical and Public Health Institute (contract agreement number 81059430). The funders had no role in study design, data collection and analysis, decision to publish, or preparation of the manuscript.

**Competing interests:** The authors declare that they have no competing interests.

## Introduction

A well-performing workforce is imperative to equitable and efficient health service delivery. Although efficiency and quality improvements of health workforce have been a major concern in the context of health sector reforms, few countries have systematically addressed performance improvements. Additionally, serious workforce shortages [1–3] are being observed across the globe that are further aggravated by poor human resource management practices, large imbalances in the geographical distribution of the health workforce [1, 4–7] and migration [1, 7–11]. It is therefore within the remit of health policy and practice to identify approaches for improving the performance of the existing health workforce.

One critical dimension of workforce performance is productivity. Productivity can be measured using benchmarks that derive from duties defined on job descriptions, against which performance indicators can afterwards be compared [12]. To assess this, work-time patterns is an important aspect of interest.

Different authors suggest to analyze time allocation patterns through various methods such as "time–motion" studies [13–15], Methods Time Measurement (MTM), Maynard Operation Sequence Technique (MOST) etc. [16–19]. The latter two are anchored in the industrial sector, mainly manufacturing or an environment with clearly defined workflow practice and typically aim to help determining and optimizing the standard time of a specific work process [16–19]. In time motion studies, frequently used in health sector, working patterns are analyzed along tasks that are divided into discrete steps and observed for the movements needed for these tasks to be completed, and measures the time that each movement takes [20]. Different techniques are used to conduct time–motion studies, but the continuous observation is known to be the most accurate one [20, 21]. During continuous observation the observer shadows the observed individual while keeping track of tasks that are being performed and records the time that each task takes [20, 22–24]. However, owing to the lack of feasibility at large scale and the high costs that this method presents, there is a preference for using alternative methods, in larger surveys, such as self-administered timesheets, retrospective interviews and data captured by computerized systems [13, 15, 20]. The current study used a time-motion approach, as our objective was most importantly to capture the overall use of work time by category of health worker, and not that of a specific task, and/or to optimize the work processes.

The limited global evidence available from previous studies on work-time patterns at primary health care (PHC) indicates that health workers (HWs) at PHC level use their work-time ineffectively with substantial time used for administrative work or simply in waiting for patients. It has been shown, for instance, that nurses of PHC services in Tanzania, Cameroon and Punjab spent 56%, 46.5% and 38% of their time respectively waiting for patients [12, 25, 26], whereas PHC doctors from different USA states spent 20–41% of their time on administrative and documentation activities [22, 27, 28]. Overall, these studies indicate that there is potential for optimizing the use of the work-time.

Albania, a southeastern European post-communist country [29], has embraced the universal health coverage policy as an essential pillar of the health system [30]. Despite having a network of 413 health centers that offer an essential package of seven main services across the country, the system seems to be inefficient among else due to a lack of professional staff, inequity in the distribution of human resources, insufficient health financing and a lack of health service management and human resource development [30, 31]. The Ministry of Health and Social Protection has acknowledged the systems' limitations and is committed to improve the quality of service delivery and strengthening PHC throughout the country. Thus, understanding the work and behavior patterns of PHC workers in Albania is of importance. At present, little is known about the time allocated to different tasks such as administration, service

provision to users and outreach activities by HWs at the PHC level in Albania and neighboring countries. Having a better understanding of how PHC workers use their working time may allow for adjusting the workforce management and ensuring more efficient care for patients.

This study assesses the work-time allocation patterns of PHC workers in two regions of Albania and to compare the patterns between different cadres and urban and rural facilities.

## Methods

### Ethics statement

Ethical clearance for the study was granted by the Ministry of Health and Social Protection of Albania on 22nd October 2019 (reference number: 5035). All HWs were informed on the objectives and methodology of the study and all participants provided written consent before study participation.

### Study design and setting

The study was conducted in two out the twelve regions of Albania. The first one, Diber, is a mountainous, rural area located in the northeast of Albania with mainly agricultural production. The second one, Fier, is located in the southeast and borders the sea, and its main economic activities are the oil industry and agriculture.

In total, these two regions host 80 health centers, out of which only 13 are urban (four in Diber and nine in Fier). Health centers are the basic unit of service provision located in towns and center of communes and to them health post-ambulatories and ambulatories are affiliated. Ambulatories were also treated as health post facilities, within the scope of this study.

### Study sampling

The time–motion study was conducted in health centers and their affiliated health post-ambulatories in the two regions corresponding in Albania to the PHC level. The following criteria had to be met by the health center for it to be eligible for study inclusion: (i) at least one medical doctor working at the health center; (ii) at least one nurse working at the health center; (iii) at least one affiliated health post-ambulatory with a nurse working at the health post-ambulatory. Eligible health centers were randomly included. In each of the two study regions, nine health centers (six rural, three urban), four rural health post ambulatories and two urban health post ambulatory were included. At the health center level, one doctor and one nurse were selected for observing work time use, while at the health post-ambulatory level, one nurse was selected for study inclusion. This resulted in an overall sample of 48 HWs.

### Data collection tool

For each of the HWs included data of his/her demographic and educational background was collected as well as selected characteristics of the health facilities he/she has been working at the time of the study.

Using the approach deployed by previous time motion studies [22–24, 32–35], an observation tool was used to observe and document key tasks of HWs and to collect and categorize activities performed. The tool reflected outlined standards of the Ministry of Health and Social Protection for doctors and nurses of PHC in Albania. The tool was pre-tested at the level of health facilities which were later not part of the study. The tool capturing work patterns was built around eight categories of activities, i.e., (i) service provision to users; (ii) administration;

(iii) continuous medical education; (iv) unproductive; (v) miscellaneous; (vi) meetings, (vii) outreach, (viii) other. The eight categories were then further broken down in 21 sub-categories and 25 subordinate sub-categories of activities, as presented in S1 Table.

## Data collection

Data collection took place from the 7th to the 28th of January 2020 using tablets, shortly before the Covid-19 pandemic hit Albania. Data were gathered by 15 observers, who received a two-day in-class training on the questionnaire and the use of the electronic data collection instrument, followed by a one day pre-test of the approach.

Conditional to presence at work, each HW was shadowed and followed-up over five consecutive days by the same observer. The time recording was conditional to the HW being at his/her workplace. In case of HW being absent, the reason for the absence was registered and no time recording was conducted. If the HW left the workplace earlier than the official duty hours, the reason for this was recorded and the observation stopped. Additionally, in many facilities nurses had their work divided in morning and afternoon shifts. During the afternoon shifts, nurses would go for home injections and visits. Due to logistics constrains of the study, no observation took place during the afternoon shifts.

The start and end of work was recorded. Additionally, the tool captured if the HW was performing facility-based or outreach activities. In the scope of this study, outreach activities included all the activities that were performed not on-site, i.e., home visits, consultation at schools, door to door visits, surveillance activities, health promotion in community, specifically (see S1 Table) and all the other main categories of activities that took place away from the facility. When the HW would leave the facility to conduct a non-site activity, the tool would first capture that an outreach activity was being performed and then the specific outreach activity that was being performed. Further, the number of patients consulted by the HWs per day was recorded.

## Data management and analysis

Data were entered using Open Data Kit (ODK) v1.25.1 and analyzed using Stata/IC (version 14.0., College Station, Texas). Data was checked for correctness and plausibility. We then applied simple descriptive statistics to analyze the data. The duration of each activity was captured in seconds. The total observed time was calculated by multiplying the total number of activities by their duration in hours. The percentage of time spent on each activity was calculated by dividing the time spent on that activity by the total observed time.

As the time recording was conditional to presence at work, we also estimated the overall work-time patterns. For doing so, we used the official duty hours per day as given by the Albanian regulations for governmental health workers, i.e., six hours and 40 minutes and six working days. To extrapolate the overall worktime, we multiplied the official duty hours for each HW by five, which stands for the number of observation days. The percentage of observed working hours was calculated by dividing the respective recorded time with the total of overall worktime. Further, the number of hours HWs lost due to absenteeism was calculated, which in this study stands for full day absences, coming late to work and leaving work earlier than official duty hours. To do so, we deducted the amount of hours observed per HW from the amount of duty hours HWs reported to work. The percentage for each absenteeism' reason was calculated by dividing the respective time by the total of overall worktime.

Four days of observation as well as the afternoon shifts could not be tracked, and they were excluded from further analyses.

**Table 1. Characteristics of the observed staff.**

| Characteristics | Doctors | | | Nurses | | | | |
|---|---|---|---|---|---|---|---|---|
| | Urban | Rural | Total | Urban health center | Rural health center | Urban health post ambulatories | Rural health post ambulatories | Total |
| Number observed | 6 | 12 | 18 | 6 | 12 | 4 | 8 | 30 |
| Average age (years) | 46 | 43.9 | 44.6 | 39 | 48.9 | 45.3 | 41.2 | 44.4 |
| % female | 100 | 50 | 66.7 | 83.3 | 83.3 | 100 | 87.5 | 86.7 |
| Average years of practice | 22 | 17.3 | 18.9 | 12.5 | 25.5 | 20.5 | 14.5 | 19.3 |
| Median of years of practice | 27.5 | 20.5 | 23.5 | 10 | 28.5 | 23.5 | 15 | 17 |

## Results

### Characteristics of health workers

The sample was made up by 18 doctors and 30 nurses. Table 1 presents the characteristics of these HWs.

Seventeen doctors reported to work 6 hours and 40 minutes daily, while one doctor worked only 6 hours daily due to a status of limited ability to work. Fifteen of the observed nurses were originally trained as general nurses, 13 as midwife-nurse, one as a medical assistant, and one for one person the professional background could not be captured. Six of the observed nurses completed a master's program in nursing, three completed a nursing course and one did another training course.

### Overall time allocation

The direct observations at health center and ambulatory / health post level allowed to track and record 76.3% of the overall working hours (1216.7 hours). In 17% of official duty hours, the HWs included were absent from the workplace. Reasons for these absences were: coming late to the workplace (4.4%), leaving earlier (4.9%) and being absent for full working days (7.7%) (Fig 1). Most of the time the absenteeism was explained by health workers by a lack of

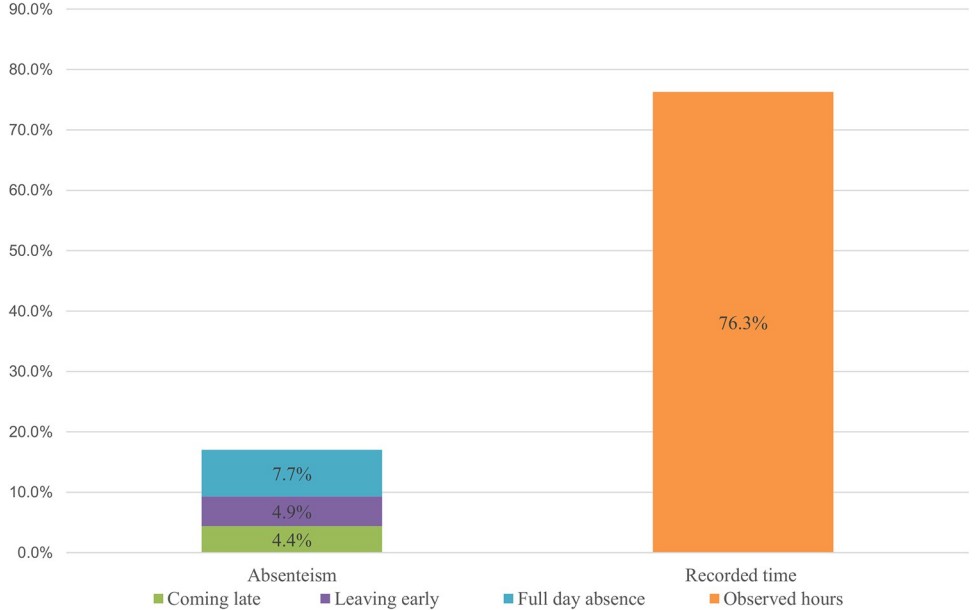

**Fig 1. Percentage of observed time and percentage of lost time (not recorded) related to working hours.**

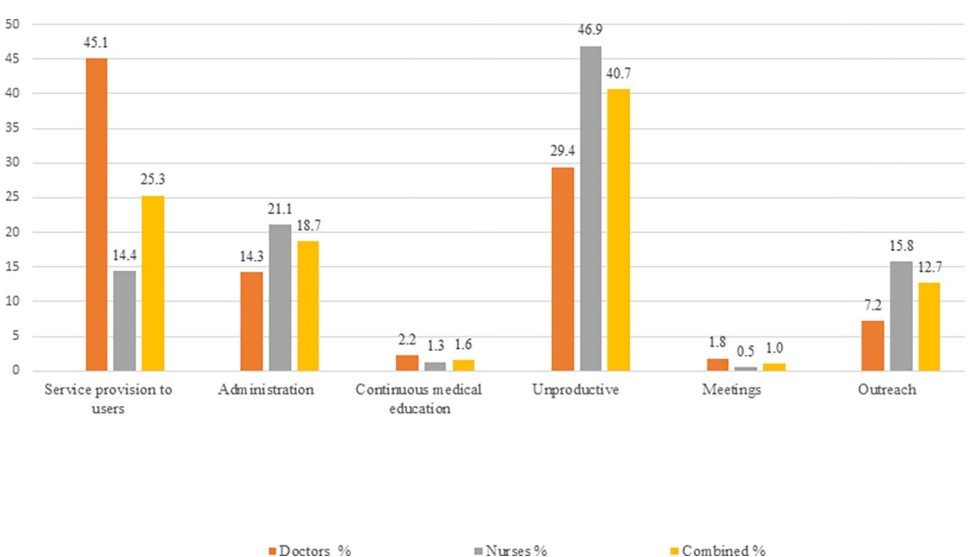

**Fig 2. Percentage of overall time allocation on categories of activities.**

transportation mean to come and get back from work, rotation and replacement within health services, or personal reasons.

HWs worked on average 5.6 hours daily (min 1.5 hours; max 7.9 hours). Over this period, they had an average of 11.4 patient consultations; doctors saw an average of 18.2 patients (min 0; max 26) per day and nurses 7.4 patients (min 0; max 28). Consequently, doctors examined an average of 3.3 patients per hour while nurses saw on average 1.3 patients per hour, excluding patient contacts during non-observed afternoon shifts.

Regarding time allocation patterns, HWs spent most of their overall working time on unproductive activities (40.7%), followed by service provision to users (25.3%), administrative tasks (18.7%), and outreach (12.7%) (Fig 2 and Table 2).

Differences in work-time allocation patterns were observed across cadres, with nurses spending more time unproductively, on administrative and on outreach activities, and consequently less time on other activities; due to smaller sample size (n = 18), the estimates for doctors are less precise than for nurses (n = 30) (see S1 Table) Both doctors and nurses spent most of their time waiting for patients (24.8% and 43.3%, respectively). Furthermore, nurses spent a substantial time on administrative tasks (21.1% versus 14.3% for doctors), mostly on filling forms for the health information system (12.1% versus 6.4% for doctors). Regarding service care provision, nurses spent 2.6% of their overall working time on medical procedures, 2.3% on preparation for patient care and 2.2% on check-up. Further, 3.6% of their time was allocated to breaks. Doctors spent most of their time on service provision. Furthermore, doctors spent 20.6% of their overall working time for administering patient forms, followed by writing prescription (11.5%) and adult and adolescent care (4.2%). Further, doctors used 4.6% of their overall working time for breaks. Both nurses and doctors spent less time on breaks than the labor regulations foresee.

Time allocation patterns during outreach corresponded to 7.2% and 15.7% of overall work time of doctors and nurses, respectively. When further breaking down this category, it was observed that doctors and nurses spent most of their time within this category on miscellaneous activities like walking/driving and/or rotation (3.4% versus 8.7%). This was followed for nurses by service provision tasks (4.4%), and for doctors by meetings (1.5%).

**Table 2. Overall time allocation on categories and sub-categories of activities by health workers category.**

| Category | Doctors | Nurses | Combined |
|---|---|---|---|
| | % | % | % |
| **Service provision to users** | **45.1** | **14.4** | **25.3** |
| Child care | 1.4 | 1 | 1.1 |
| Woman and reproductive health care | 0.1 | 0.2 | 0.2 |
| Adult and adolescent care | 4.2 | 0.3 | 1.7 |
| Clinical consultation | 1.8 | 0.2 | 0.8 |
| Check up | 0.1 | 2.2 | 1.5 |
| Assistance with services | 0.3 | 0.1 | 0.2 |
| Medical procedures | 2.9 | 2.6 | 2.7 |
| Other patient consultation | 1.3 | 0.4 | 0.8 |
| Health promotion | 0.4 | 0.1 | 0.2 |
| Clinical orientation in the reception | 0.1 | 0.3 | 0.2 |
| Writing prescriptions | 11.5 | 0 | 4.1 |
| Preparation for patient care | 0.4 | 2.3 | 1.6 |
| Patient administration | 20.6 | 4.7 | 10.3 |
| **Administration** | **14.3** | **21.1** | **18.7** |
| Health information system | 6.4 | 12.1 | 10.1 |
| Other administration | 7.9 | 9 | 8.6 |
| **Continuous medical education** | **2.2** | **1.3** | **1.6** |
| Professional reading | 2.2 | 1.3 | 1.6 |
| Training | 0 | 0 | 0 |
| **Unproductive** | **29.4** | **46.9** | **40.7** |
| Waiting | 24.8 | 43.3 | 36.8 |
| Break | 4.6 | 3.6 | 3.9 |
| **Meetings** | **1.8** | **0.5** | **1** |
| **Outreach** | **7.2** | **15.8** | **12.7** |
| Service provision to users | 0.7 | 4.4 | 3.1 |
| Administration | 0.4 | 0.6 | 0.5 |
| Continuous medical education | 1 | 0 | 0.4 |
| Unproductive | 0.1 | 0.8 | 0.6 |
| Miscellaneous | 3.4 | 8.7 | 6.8 |
| Other | 0 | 1 | 0.6 |
| Meetings | 1.5 | 0.2 | 0.7 |

The categories of activities under the bold 'Outreach' activity, capture the amount of time that was spent in each of them, respectively, while these activities were being performed not on-site.

### Work-time allocation patterns at rural and urban health facilities

Doctors in both urban and rural health centers spent most of their time with service provision activities (54.0% versus 41.2%), followed by unproductive activities (30% versus 29.3%) and administrative tasks (17.3% versus 9.7%) (Table 3) (see confidence intervals in S1 Table). While for a number of categories differences were small, doctors working in urban health centers spent more time for service provision than doctors of rural health centers, the latter spent more time on–most notably–miscellaneous activities (5.3% versus 0.2%) and outreach (10.7% of their overall working time compared with 1.2% for doctors in urban settings) (Table 4).

Nurses positioned in rural health centers spent more time than those working in urban settings on–most notably – miscellaneous activities (6.6% versus 4.4%), but less on

**Table 3. Percentage of overall time allocation (including outreach time) by HWs category at rural and urban settings.**

| Health worker category | Doctors (%) | | Nurses (%) | | | |
|---|---|---|---|---|---|---|
| Category | Rural health center | Urban health center | Rural health center | Rural health post ambulatories | Urban health center | Urban health post ambulatories |
| Service provision to users | 41.2 | 54 | 19.5 | 18.7 | 18.8 | 18 |
| Administration | 17.3 | 9.7 | 20.6 | 15.5 | 26.1 | 28 |
| Continuous Medical Education | 3.5 | 2.8 | 2.1 | 0.6 | 1.2 | 0.5 |
| Unproductive | 29.3 | 30 | 49 | 50.2 | 48.1 | 40.7 |
| Miscellaneous | 5.3 | 0.2 | 6.6 | 12.9 | 4.4 | 12.5 |
| Other | 0 | 0 | 1.6 | 1.2 | 0.1 | 0 |
| Meetings | 3.4 | 3.3 | 0.6 | 0.9 | 1.2 | 0.3 |

administration (20.6% versus 26.1%) and meetings (0.6% versus 1.2%) (Table 3) (see confidence intervals in S1 Table).

Nurses of rural health post-ambulatories spent more time compared to those working in urban areas on–most notably – unproductive activities (50.2% versus 40.7%), but less on administration activities (15.5% versus 28.0%) (Table 3). Furthermore, nurses in rural settings spent more time on outreach compared to nurses in urban ones (Table 4).

## Discussion

The results of this study indicate that doctors and nurses of PHC facilities spend the largest proportion of their time on unproductive activities (40.7%), one fourth on service provision (25.3%) and 18.7% on administrative activities. The work-time allocation patterns among the professionals assessed differ, with nurses spending more time on unproductive, administrative and outreach activities and less on all other activities compared with doctors.

Previous studies conducted in PHC facilities in Sweden, as well as other settings, such as surgical centers, nursing homes and community centers in USA and Brazil have also reported that nurses spend a substantial proportion of the work time on administrative activities [36–41]. The effort required for the administrative tasks in this study is illustrated by the fact that there are 58 registers and forms to be filled in by staff of PHC facilities [42] often asking for overlapping information. One reason for this duplicated effort is that the HWs of PHC report separately to two different institutions: the regional entities of the Compulsory Health Care Insurance Fund and the Local Health Care Units of the Health Operator. Moreover, most of

**Table 4. Percentage of overall time allocation during outreach by HWs category at rural and urban settings.**

| Time allocation at rural and urban settings by health worker category while on outreach | | | | | | |
|---|---|---|---|---|---|---|
| Health worker category | Doctors (%) | | Nurses (%) | | | |
| Category | Rural health center | Urban health center | Rural health center | Rural health post ambulatories | Urban health center | Urban health post ambulatories |
| Service provision to users | 13.3 | 0 | 31.2 | 27.9 | 20.5 | 28.2 |
| Administration | 2.9 | 3.5 | 1.3 | 2.6 | 17.4 | 2.9 |
| Continuous Medical Education | 15.2 | 0 | 0.7 | 0 | 0.6 | 0 |
| Unproductive | 2 | 0 | 1.7 | 8.4 | 11.7 | 0.9 |
| Miscellaneous | 49.2 | 16.7 | 52.4 | 52.6 | 48.6 | 68.1 |
| Other | 0 | 0 | 12.7 | 5.1 | 1.3 | 0 |
| Meetings | 17.5 | 79.8 | 0 | 3.4 | 0 | 0 |

the documentation is paper based. Improving the health information practices for PHC workers, through better coordination between central institutions, more investment in digitalization [15, 27, 29], and training of staff on gaining skills on such digitalized health services could improve the efficiency of PHC operations. However, these entail a redesign of reporting requirements, financial investments in training of the HWs, equipping the health centers with computers and ensuring stable access to internet [15].

The substantial share of work time spent on unproductive activities observed in this study (41%) is in line with previous work. Two recent reports, e.g., have reached similar conclusions revealing that nurses of PHC in Albania are under-used [see S1 and S2 Texts]. In Tanzania, nurses in reproductive and child health clinics were observed to work productively only at 57% of the time present at the facility [12], whereas in Punjab the auxiliary nurse midwives spent only 62% of their time productively [25]. In Cameroon, the proportion of time that nurses spent unproductively reached 73% [26]. As for the doctors, the amount of time that they allocated to unproductive activities in the current study was 20%-24% more than what is reported as personal time among PHC doctors, internists and hospitalists in other studies in USA and Germany [22, 23, 28, 43].

Our study showed that doctors spent most of their time on service provision (45%), a bulk of which is dedicated to administering patients' forms (21%), leaving 24% for direct patient contact. Studies in the USA and Germany focusing on primary care doctors, hospitalist, internist and hospitalist have reported a high amount of time spent on patient documentation, from 21% to 49% [22, 23, 27, 28, 43]. Additionally, in these studies the observed doctors spent between 17% and 55% on direct patient care.

In the current study doctors spent more time than nurses on service provision to patients (45% versus 14%), as well as direct patient contact (24.5% versus 9.7%). As in other settings, nurses do not have the authority and competencies as well as the self-confidence to conduct certain activities, especially the ones involving more complex skills and clinical reasoning [44] [see S1 and S2 Texts]. Indeed, Albania, as other countries [see S2 Text], has a long tradition of doctor-centered health care with limited scope of nurse practice, upon whom a secretarial role is imposed. In addition, the Basic Package of Care despite outlining the required skills for family doctors and nurses at PHC facilities, does not give a clear description of their roles. Clear role descriptions establish a ground for respect and trust among health providers, thus, facilitate and improve teamwork [45] and they are associated with improved performance [5]. It is indeed an objective of the upcoming Strategy on the Development of PHC in Albania 2020–2025 to reassess, reformat and improve PHC professionals' job descriptions [see S1 Text].

In the light of the results of this study, better management of PHC services in terms of planning of HWs' activities is needed. It is necessary to train the HWs in work-time organization [see S1 Text], as this is an important skill that PHC staff should develop in order to use their work time efficiently. A future study, using the MOST method, could determine the processes of a clinical consultation and provide insights to optimize these processes [17–19]. Further, implementing an appointment system for patients, namely those with a chronic condition regularly visiting a PHC service, could contribute to optimize the use of work time and reduce time periods where HWs wait for patients.

Additionally, several other measures to improve efficiency could be taken i.e. nurses' empowerment in PHC through training and education programs, nurses' extension area of work by granting them more responsibility and more equilibrated shares between doctors and nurses [45–47] and evolvement of nurses' roles by adapting the legislation and education structures [48].

We assumed that 17% of the working hours were accounted for as absenteeism in the workplace, which was caused by, e.g., late arrival to the workplace, leaving early and being absent for full working days. Various reports have described HWs' absenteeism in the workplace. A

multicountry survey, conducted in several low and middle Income countries reported rates of absenteeism of PHC workers ranging from 25% to 40% [49]. According to this survey, factors such as irregular supervision, inadequate working conditions, logistics to get to work, rural location and absence of staff houses influenced the high rates of absenteeism. To reduce absenteeism in workplace, Manzi et al. suggested improvement of HWs' management through adequate supervision, improved supplies of essential goods and on the job training [12].

## Study limitations

This study had several limitations. To begin with, time motion studies impose the risk of the Hawthorne effect [50]. In the current study, HWs were shadowed by an observer, which may have altered their working behavior and reduced the level of absenteeism and unproductive work. If so, the main conclusion in terms of work-time use would however not altered and patterns in terms of inefficient use of work-time would rather look more unfavorable. As an attempt to decrease the effect of the observer by allowing them to become acclimated to the presence of the observer HWs were observed for several days.

Second, for logistical reasons it was impossible to obtain the observations in the afternoon hours while some of the nurses conducted home visits, for example administering injections to patients; resulting in lower registered time allocated to outreach activities.

Third, we believe that similar work-time patterns as observed in the chosen facilities in Fier and Diber are found across both regions. Nevertheless, we did not account for clustering effects during the analysis, thus the findings should be interpreted cautiously. Further, these findings should not be generalized in a whole country context as in urban regions such as Tirana or Durres, the work-time allocations patterns might differ substantially.

Fourth, we only assessed the time allocation of health providers at one point during the year, in winter time, and therefore this may not be representative of yearly utilization patterns. In other warmer periods of the year utilization rates of health centers may be higher leading then also to altered time allocation patterns i.e.: more time allocated to service provision and administration, leading eventually to less unproductive time.

In spite of its limitations, this study offers valuable insight on work-time patterns of doctors and nurses in urban and rural settings, health centers and health posts in the two regions of Albania.

## Conclusion

This study indicates that HWs devote a substantial amount of work time to unproductive, service provision to users and administrative activities. Consequently, there is potential to make better use of the available working time of doctors and nurses working at PHC level in Albania. Also, we found that work-time patterns differ among cadres, with nurses spending less time, compared to doctors, on all activities except for unproductive and outreach activities. Additionally, our findings revealed that work-time allocation patterns were similar between urban and rural settings.

Our results suggest that there is potential for efficiency gains in PHC workers' time allocation. Improved health information system practices, appointment system establishment, empowered nurses and clear share of responsibilities might contribute to more efficient use of time.

## Supporting information

**S1 Table. Categories of activities to observe.**
(DOCX)

**S2 Table. Overall time allocation and 95% confidence intervals on categories of activities by health workers.**
(DOCX)

**S3 Table. Percentage of overall time allocation (including outreach time) and 95% confidence intervals by doctors at rural and urban settings.**
(DOCX)

**S4 Table. Percentage of overall time allocation (including outreach time) and 95% confidence intervals by nurses at rural and urban settings.**
(DOCX)

**S1 Text. Analysis of the context in the view of development and implementation of new job profiles and roles for family nurses in Albania.**
(DOCX)

**S2 Text. Nurse profiles and master in family nursing.**
(DOCX)

## Acknowledgments

We would like to thank Dr Besim Nuri and the team of the "Health for All Project" (Projekti "Shëndet për të Gjithë") for their support on the field work. Special thanks to Manuela Murthi, Arti Cicolli and all of the observers for their outstanding contributions. Additionally, we are extremely thankful for the valuable support of Dr Aurelio Di Pasquale and Dr Marek Kwiat-kowski. Finally, we would like to thank the family doctors and nurses of Diber and Fier for participating in our study.

## Author Contributions

**Conceptualization:** Altiona Muho, Kaspar Wyss.

**Data curation:** Altiona Muho.

**Formal analysis:** Altiona Muho.

**Funding acquisition:** Kaspar Wyss.

**Methodology:** Altiona Muho, Altina Peshkatari, Kaspar Wyss.

**Project administration:** Altiona Muho, Altina Peshkatari.

**Software:** Altiona Muho.

**Supervision:** Altiona Muho.

**Writing – original draft:** Altiona Muho.

**Writing – review & editing:** Altina Peshkatari, Kaspar Wyss.

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
