## [Decision Letter · Decision Letter 0]

1 Jul 2022

PONE-D-22-01564Work time allocation at primary health care level in two regions of AlbaniaPLOS ONE

Dear Dr. Muho,

Thank you for submitting your manuscript to PLOS ONE. After careful consideration, we feel that it has merit but does not fully meet PLOS ONE’s publication criteria as it currently stands. Therefore, we invite you to submit a revised version of the manuscript that addresses the points raised during the review process.

 Please see the comments of one reviewer below. Please note that we have only been able to secure a single reviewer to assess your manuscript. We are issuing a decision on your manuscript at this point to prevent further delays in the evaluation of your manuscript. Please be aware that the editor who handles your revised manuscript might find it necessary to invite additional reviewers to assess this work once the revised manuscript is submitted. However, we will aim to proceed on the basis of this single review if possible.  Please submit your revised manuscript by Aug 14 2022 11:59PM. If you will need more time than this to complete your revisions, please reply to this message or contact the journal office at plosone@plos.org. Please include the following items when submitting your revised manuscript:A rebuttal letter that responds to each point raised by the academic editor and reviewer(s). You should upload this letter as a separate file labeled 'Response to Reviewers'.A marked-up copy of your manuscript that highlights changes made to the original version. You should upload this as a separate file labeled 'Revised Manuscript with Track Changes'.An unmarked version of your revised paper without tracked changes. You should upload this as a separate file labeled 'Manuscript'.If applicable, we recommend that you deposit your laboratory protocols in protocols.io to enhance the reproducibility of your results. Protocols.io assigns your protocol its own identifier (DOI) so that it can be cited independently in the future. For instructions see: https://journals.plos.org/plosone/s/submission-guidelines#loc-laboratory-protocols. Additionally, PLOS ONE offers an option for publishing peer-reviewed Lab Protocol articles, which describe protocols hosted on protocols.io. Read more information on sharing protocols at https://plos.org/protocols?utm_medium=editorial-email&utm_source=authorletters&utm_campaign=protocols.

We look forward to receiving your revised manuscript.

Kind regards,

Hanna Landenmark

Staff Editor

PLOS ONE

Journal Requirements:

Reviewers' comments:

Reviewer's Responses to Questions

**Comments to the Author**

1. Is the manuscript technically sound, and do the data support the conclusions?

Reviewer #1: Partly

2. Has the statistical analysis been performed appropriately and rigorously? 

Reviewer #1: Yes

3. Have the authors made all data underlying the findings in their manuscript fully available?

Reviewer #1: Yes

4. Is the manuscript presented in an intelligible fashion and written in standard English?

Reviewer #1: Yes

5. Review Comments to the Author

Reviewer #1: The objective of this study is to assess the work-time allocation patterns of PHC workers in two regions of Albania and to compare the patterns between doctors and nurses and between urban and rural facilities. This study offers valuable insight on work-time patterns of doctors and nurses in urban and rural settings, in the two regions of Albania where the research took place.

I have the following comments, questions and suggestions:

1. For some activities the differences between rural and urban facilities are rather small, but the authors report that one group spends more time on the activity than the other group. For example: line 230: “meetings (3.4% versus 3.3%)”. It is recommended that the size of the differences is also taken into account.

2. Related to the previous comment, the observations are based on a sample. Is it not possible to calculate confidence intervals around the point estimates?

3. The authors mention several limitations. It would be interesting to discuss more the implications of the limitations for the results. To what extent can the limitations influence the result?

4. In each of the two study regions, nine health centers were included and at the health center level, one doctor, one nurse and one nurse at the affiliated health post-ambulatory. This should result in 54 HWs (2*9*3=54). However only 48 HWs participated in the study. Could be clarified which category of HWs are missing and why? I suggest to split in table 1 the category nurses in 'nurses at the health center' and 'nurses located at the affiliated health post-ambulatories'.

5. The added value of figure 2 escapes me since this information can also be found in table 2. Furthermore, the legend of figure 2 stated a category that cannot be found in the figure.

6. Table 2 is very important, however, the presentation of the categories is not clear. For example, the categories ‘other’, ‘meetings’, ‘administration’ are mentioned several times.

6. PLOS authors have the option to publish the peer review history of their article (what does this mean?). If published, this will include your full peer review and any attached files.

Reviewer #1: **Yes: **Wim Peersman

---

## [Author Response · Author response to Decision Letter 0]

29 Jul 2022

Response to the first reviewer's comment: We acknowledge this comment and agree. Consequently reworded the respective section. We indicate now that for a number of categories differences between urban and rural being small, but also indicate those categories where differences are being observed (lines 234-242). 

Response to the second reviewer's comment: Thank you for bringing this important point to our attention. We now have calculated the confidence intervals for each of the categories of activities for doctors and nurses, in urban and rural settings, respectively. So not to overload the tables and to keep them well understandable, the confidence intervals are presented into three new tables as supporting information (reflected in the lines 508-513) under the titles: S4 Table, S5 Table, S6 Table. In text citation referring to the confidence intervals in supporting information are added (lines 212-213, 233, 241-242). 

Response to the third reviewer's comment: We agree with the reviewer. Consequently we updated the limitation section accordingly (lines 326-327, 332-333, 337, 341-342) and hope it meets the reviewer’s requests. 

Response to the fourth reviewer's comment: Thank you for this valid point and for the suggestion. We updated in the revised version of the manuscript table 1 and the narrative of the sampling so to better clarify why there is only 48 HWs and not 54. Indeed, there are only two urban health post ambulatories sampled per region, thus there are four nurses less from these facilities (changes in the lines 115-116). Further, we split the category nurses in four sub-categories in Table 1 i.e.: ‘nurses at urban health center’, ‘nurses at rural health center’, ‘nurses at urban health post-ambulatories’ and ‘nurses at rural health post-ambulatories’ (line 179). 

Response to the fifth reviewer's comment: We would like to acknowledge reviewer’s observation about the similarities that figure 2 and table 2 present. However, we believe that figure 2 adds to the clarity and visibility of our manuscript, making it easy for the reader to interpret the data. We hope it being acceptable to keep both tables included. Further, apologies for the typo in the legend. The category of activity from the legend of figure 2 the reviewer refers to in the comment, has been replaced accordingly (from ‘Direct patient care’ to ‘Service provision to users’). 

Response to the sixth reviewer's comment: We thank the reviewer for this observation. We believe that this information can be found under the ‘Data collection section’ and we tried to further clarify it by adding an explanatory sentence (see lines 148-150). Further, we placed a footnote under Table 2 (lines 207-209) and we hope it meets the reviewer’s requests.

Response to the first journal's requirement: Requirement acknowledged. We reviewed our reference list and there is no article that has been retracted so far. However, we did noticed that in the first submission we had missed to number one of the references (reference number 41, line 483). Subsequently, all the references after this one in the reference list, increased their number by one. Only one change was made in the text due to this (line 311). 

Response to the second journal's requirement: We thank the editor for bringing this point to our attention. We added a header to table 4 (line 248) in order to make the reading of the table more understandable.

---

## [Decision Letter · Decision Letter 1]

12 Sep 2022

PONE-D-22-01564R1Work time allocation at primary health care level in two regions of AlbaniaPLOS ONE

Dear Dr. Muho,

Thank you for submitting your manuscript to PLOS ONE. After careful consideration, we feel that it has merit but does not fully meet PLOS ONE’s publication criteria as it currently stands. Therefore, we invite you to submit a revised version of the manuscript that addresses the points raised during the review process.

We look forward to receiving your revised manuscript.

Kind regards,

Dragan Pamucar

Academic Editor

PLOS ONE

Journal Requirements:

Reviewers' comments:

Reviewer's Responses to Questions

**Comments to the Author**

1. If the authors have adequately addressed your comments raised in a previous round of review and you feel that this manuscript is now acceptable for publication, you may indicate that here to bypass the “Comments to the Author” section, enter your conflict of interest statement in the “Confidential to Editor” section, and submit your "Accept" recommendation.

Reviewer #1: (No Response)

Reviewer #2: All comments have been addressed

2. Is the manuscript technically sound, and do the data support the conclusions?

Reviewer #1: Yes

Reviewer #2: Yes

3. Has the statistical analysis been performed appropriately and rigorously? 

Reviewer #1: I Don't Know

Reviewer #2: Yes

4. Have the authors made all data underlying the findings in their manuscript fully available?

Reviewer #1: Yes

Reviewer #2: No

5. Is the manuscript presented in an intelligible fashion and written in standard English?

Reviewer #1: Yes

Reviewer #2: Yes

6. Review Comments to the Author

Reviewer #1: My comments have only been partially answered.

Comment 2 (concerning the confidence intervals): in the tables S4, S5, S6, the point estimate is a percentage and the confidence intervals are proportions. It would be more clear to report the confidence intervals also as a percentage.

Comment 4 (concerning the number of HWs that participated): table 1 is very clear, but is it possible that the narrative is not entirely correct?

Comment 5 (concerning figure 2): the legend of figure 2 stated a category (‘main category’) that cannot be found in the figure.

Reviewer #2: This Manuscript is good and quite innovative. It clears the clarity of the reader. It is well structured and well written. The author does an excellent job of presenting a highly technical and complicated process in an easy-to-understand manner.

Although here are few of the observations:

1. To have an unbiased view in the paper, there should be some discussions of the various other work measurement methods like MOST, MTM-1, etc.

2. The discussion and conclusions should be extended with more future work discussion and comparison of related literature findings with above methods.

3. Justify the reason for choosing the proposed methodology.

7. PLOS authors have the option to publish the peer review history of their article (what does this mean?). If published, this will include your full peer review and any attached files.

Reviewer #1: **Yes: **Wim Peersman

Reviewer #2: No

---

## [Author Response · Author response to Decision Letter 1]

24 Sep 2022

Response to reviewer 1: 

1- Thank you for bringing this point to our attention. We have now changed the confidence intervals in the table S4, S5, S6, from proportions to percentages. 

2-We agree with the reviewer. Consequently we updated the study sampling section accordingly (line 122) and hope it meets the reviewer’s requests.

3- We would like to acknowledge reviewer’s observation on the difference between figure 2 and its legend. The typo in the legend has been removed. 

Response to reviewer 2: 

We thank the reviewer for recognizing the innovation and the clarity of the manuscript.

1- The concern of the reviewer is well acknowledged. Accordingly we have extended introduction section and provide reference to the methods for analyzing work processes (MOST and MTM), (lines 60-65). We also provide a rational why these methods were not the most suitable for addressing the research questions.

2- Thank you for bringing this aspect to our attention. The discussion has been updated accordingly, providing future work recommendations (lines 313-314).

3- This a valid point and thus thanks for raising this point. We have added a rational why a time motion method was considered the best approach for analyzing worktime allocation patterns at PHC level (lines 74-76). 

Journal requirements response: 

1- Requirement acknowledged. We reviewed our reference list and there is no article that has been retracted so far. 

Additionally, to compile with the second reviewer’s comments about presenting other methods applied for studying work processes, we added four new references supporting the updated sections (references number 16, 17, 18, 19, lines 422-429). Subsequently, all the references after these ones in the reference list, increased their number by four. Changes were made throughout the text due to this (lines 62, 64, 67, 69, 71, 74, 82, 83, 85, 86, 90, 130, 270-272, 278, 287, 288, 291, 296, 301, 307, 314, 321, 322, 327, 335).

---

## [Decision Letter · Decision Letter 2]

2 Oct 2022

Work time allocation at primary health care level in two regions of Albania

PONE-D-22-01564R2

Dear Dr. Muho,

We’re pleased to inform you that your manuscript has been judged scientifically suitable for publication and will be formally accepted for publication once it meets all outstanding technical requirements.

Kind regards,

Dragan Pamucar

Academic Editor

PLOS ONE

Additional Editor Comments (optional):

Reviewers' comments:

Reviewer's Responses to Questions

**Comments to the Author**

1. If the authors have adequately addressed your comments raised in a previous round of review and you feel that this manuscript is now acceptable for publication, you may indicate that here to bypass the “Comments to the Author” section, enter your conflict of interest statement in the “Confidential to Editor” section, and submit your "Accept" recommendation.

Reviewer #1: All comments have been addressed

Reviewer #2: All comments have been addressed

2. Is the manuscript technically sound, and do the data support the conclusions?

Reviewer #1: Yes

Reviewer #2: Yes

3. Has the statistical analysis been performed appropriately and rigorously? 

Reviewer #1: I Don't Know

Reviewer #2: Yes

4. Have the authors made all data underlying the findings in their manuscript fully available?

Reviewer #1: Yes

Reviewer #2: Yes

5. Is the manuscript presented in an intelligible fashion and written in standard English?

Reviewer #1: Yes

Reviewer #2: Yes

6. Review Comments to the Author

Reviewer #1: (No Response)

Reviewer #2: Thanks for making changes. All comments have been addressed by the authors in this version of manuscript.

7. PLOS authors have the option to publish the peer review history of their article (what does this mean?). If published, this will include your full peer review and any attached files.

Reviewer #1: **Yes: **Wim Peersman

Reviewer #2: No

---

## [Editor Report · Acceptance letter]

5 Oct 2022

PONE-D-22-01564R2 

Work time allocation at primary health care level in two regions of Albania. 

Dear Dr. Muho:

I'm pleased to inform you that your manuscript has been deemed suitable for publication in PLOS ONE. Congratulations! Your manuscript is now with our production department. 

Kind regards, 

on behalf of

Dr. Dragan Pamucar 

Academic Editor

PLOS ONE